# Faster and Scalable Algorithms for Densest Subgraph and Decomposition

**Harb, Elfarouk**[*]
eyharb2@illinois.edu
University of Illinois at Urbana-Champaign

**Quanrud, Kent**[†]
krq@purdue.edu
Purdue University

**Chekuri, Chandra**[‡]
chekuri@illinois.edu
University of Illinois at Urbana-Champaign

## Abstract

We study the densest subgraph problem (DSG) and the densest subgraph local decomposition problem (DSG-LD) in undirected graphs. We also consider supermodular generalizations of these problems. For large scale graphs simple iterative algorithms perform much better in practice than theoretically fast algorithms based on network-flow or LP solvers. Boob *et al.* [1] recently gave a fast iterative algorithm called GREEDY++ for DSG. It was shown in [2] that it converges to a $(1 - \epsilon)$ relative approximation to the optimum density in $O(\frac{1}{\epsilon^2} \frac{\Delta(G)}{\lambda^*})$ iterations where $\Delta(G)$ is the maximum degree and $\lambda^*$ is the optimum density. Danisch *et al.* [3] gave an iterative algorithm based on the Frank-Wolfe algorithm for DSG-LD that takes $O(\frac{m\Delta(G)}{\epsilon^2})$ iterations to converge to an $\epsilon$-additive approximate local decomposition vector $\hat{b}$, where $m$ is number of edges in the graph.

In this paper we give a new iterative algorithm for both problems that takes at most $O(\frac{\sqrt{m\Delta(G)}}{\epsilon})$ iterations to converge to an $\epsilon$-additive approximate local decomposition vector; each iteration can be implemented in $O(m)$ time. We describe a fractional peeling technique which has strong empirical performance as well as theoretical guarantees. The algorithm is scalable and simple, and can be applied to graphs with hundreds of millions of edges. We test our algorithm on real and synthetic data sets and show that it provides a significant benefit over previous algorithms. The algorithm and analysis extends to hypergraphs.

## 1. Introduction

The densest subgraph problem (DSG) is a classical problem in combinatorial optimization and has many real world applications in data mining, network analysis, and machine learning. The input for DSG is an undirected graph $G = (V, E)$ with $m = |E|$ and $n = |V|$. The goal is to return a subset $S \subseteq V$ that maximizes $\frac{|E(S)|}{|S|}$ where $E(S) = \{\{u, v\} \in E : u, v \in S\}$ is the set of edges with both end points in $S$. DSG has a variety of applications in which dense subgraphs reveal important information about the underlying network such as communities. One can view it as a subroutine in

---

[*]Supported in part by NSF grant CCF-2028861.

[†]Supported in part by NSF grant CCF-2129816.

[‡]Supported in part by NSF grants CCF-2028861 and CCF-1910149.

We thank Professors Sariel Har-Peled and Ruoyu Sun from the University of Illinois Urbana-Champaign for fruitful discussions in private correspondences.

an unsupervised clustering procedure. It is a canonical problem in the broad area of dense subgraph discovery which has seen many developments and applications in the past two decades. We point the reader's attention to a (non-exhaustive) list of recent, and some not so recent, important work and the pointers therein [4, 5, 6, 7, 1, 3, 8, 9, 10, 11, 12, 13, 14, 15, 16, 17, 18, 19, 20, 21, 22, 23]. Constrained versions of DSG such as the densest $k$-subgraph problem DKSG, densest at most $k$-subgraph, and densest at least $k$-subgraph DALKSG are also well-studied and many of these are NP-Hard. See [24, 25, 26, 10, 6, 27] for some positive and negative results on approximation algorithms.

One of the key advantages of DSG is its polynomial-time solvability. The first polynomial-time algorithms were due to Goldberg [4] and Picard and Queuranne [28]. The decision version of DSG is the following: given $G$ and a rational number $\lambda$, is the density in $G$ at least $\lambda$? The algorithms in [4, 28] construct an auxiliary directed flow-network $H$ such that the maximum flow in $H$ allows one to answer the decision version. Binary search over $\lambda$ leads to the final algorithm. Maximum flow is a powerful algorithmic subroutine, however, it is not a practical or scalable algorithm for modern graph data sets with millions and even billions of vertices and edges. For example, authors from [1] noted that the Goldberg's maximum flow algorithm failed on many large scale graphs even though they used a highly optimized maximum flow library. Charikar [5] described a linear programming formulation for DSG that gives an exact solution and has $O(|E|)$ variables and constraints. LP solvers are also unsuitable for large data sets due to memory limitations among others. Charikar [5] described a very simple $\frac{1}{2}$-approximation algorithm for DSG known as the GREEDY or PEELING algorithm. The algorithm creates an ordering of the vertices as follows. The first vertex $v_1$ is the one with the smallest degree in $G$ (ties broken arbitrarily). It selects $v_2$ to be the smallest degree vertex in $G - v_1$. Letting $G^i$ be the graph after removing $v_1, v_2, \ldots, v_{i-1}$ (with $G^0 = G$), the algorithm returns the graph among $G^0, \ldots, G^n$ with the highest density. The algorithm can be implemented in $O(m)$ time. The ordering created by the algorithm is the same as the one to compute a $k$-core decomposition of a graph — this is a well-studied graph decomposition procedure with several applications [29] . The simplicity and the efficiency of the Greedy algorithm, and its approximation guarantees, has led to its adoption for a number of other density measures.

Despite Greedy's advantages, its worst-case approximation guarantee is only $\frac{1}{2}$, and is worse for other density measures. The goal is to develop algorithms that obtain a $(1-\varepsilon)$ relative approximation for a given parameter $\varepsilon \in (0, 1)$ while also being scalable to large graphs. One approach to obtain such algorithms is via the dual of Charikar's LP relaxation. It is a mixed packing and covering LP for a given guess of the optimal value. Such LPs can be approximately solved via iterative methods such as the multiplicative weight updates (MWU) or other methods based on convex optimization. Bahmani, Goel and Munagala [30] applied this methodology to obtain an algorithm that yields a $(1-\varepsilon)$-approximation in $\tilde{O}(m/\varepsilon^2)$-time. Boob, Sawlani and Wang [31] described an algorithm that yields a $(1 - \varepsilon)$-approximation in $\tilde{O}(m\Delta(G)/\varepsilon)$-time, where $\Delta(G)$ is the maximum degree in $G$. More recently Chekuri, Quanrud and Torres [2] obtained a $(1 - \varepsilon)$-approximation in $\tilde{O}(m/\varepsilon)$-time via approximate flow techniques. Some of these nice theoretical developments have not yet led to practically useful algorithms for large scale graphs. In a different direction, Boob *et al.* [1] described a fast iterative algorithm called GREEDY++ which builds on the Greedy algorithm and insights from the LP relaxation. It does extremely well in experiments, and the authors conjectured that it yields a $(1 - \epsilon)$ relative approximation in $O(\frac{1}{\epsilon^2})$ iterations and each iteration can be implemented in $O(m)$ time. Chekuri *et al.* [2] proved that GREEDY++ converges to a $(1 - \epsilon)$-approximation in $O(\frac{\Delta(G)}{\lambda^* \epsilon^2})$ iterations where $\lambda^*$ is the optimum density. This gives evidence of the theoretical soundness of the algorithm.

Our goal in this paper is to develop new and scalable algorithms that outperform GREEDY++ while having strong theoretical guarantees. In addition, we are interested in an algorithm that finds an approximate *dense subgraph decomposition* which gives information on the density structure of the graph and is different from another such decomposition, namely the $k$-core decomposition. Further, as shown in previous work [32, 33, 2], a dense subgraph decomposition allows one to compactly represent approximate solutions to densest subgraphs of different sizes; in particular the representation yields a 2-approximation for densest at least $k$-subgraph for any given $k$ [32, 2].

**Dense subgraph decomposition.** One can show that every graph $G = (V, E)$ admits a certain structured nested decomposition based on decreasing density. In particular, the vertex set $V$ can partitioned into $S_1, S_2, \ldots, S_k$ such that $S_1$ is the unique maximal densest subgraph in $G$, and $S_i$ is the unique maximal densest set with respect to $S_1 \cup \ldots \cup S_{i-1}$ (a formal definition is deferred to

Section 3). For each vertex $v$ we let $\lambda_v$ denote the density of the set containing $v$. The existence of such a decomposition follows from the supermodularity properties of the function $f(S) = |E(S)|$, and is implicitly known from past work on submodular functions (in particular, the classical result of Fujishige [34]). For graphs this decomposition was rediscovered by Tatti and Gionis [35, 12] under the name of locally-dense decomposition. Tatti showed that one can compute the exact decomposition via $n$ maximum flow computations. Danisch *et al.* [3] showed that computing the $\lambda_v$ values can be cast as solving a quadratic program that extends the LP of Charikar, and applied the Frank-Wolfe algorithm. They showed that their algorithm needs $O(\frac{|E|\Delta(G)}{\epsilon^2})$ iterations (each taking $O(|E|)$ time) to converge to an $\varepsilon$-approximate vector. Their work shows that the Frank-Wolfe algorithm leads to a good approximation to the densest subgraph in a relatively small number of iterations based on experiments. However, they do not describe a systematic way to extract a dense subgraph decomposition from the approximate vector with provable guarantees. See Figure 6.4 in Appendix 6.1 for a table summarizing known results. For DSG-LD, GREEDY++ is not proven to converge to the optimal dense decomposition load vector [4]. The FRANK-WOLFE based algorithm takes $O(\frac{m\Delta(G)}{\epsilon^2})$ iterations, each taking $O(m)$ time to converge to a $(1-\epsilon)$ dense decomposition. Our algorithm requires $O(\frac{\sqrt{m\Delta(G)}}{\epsilon})$ iterations, each taking $O(m)$ time, to converge to a $(1-\epsilon)$ dense decomposition vector. Finally, the multiplicative weight update algorithm takes $O(\frac{m\Delta(G)}{\epsilon^2})$ iterations, each taking $O(m)$ time. We note that the multiplicative weight update algorithm has several variants, and the one we implement is discussed in Section 5.5. Except for GREEDY++ due to the peeling nature of the algorithm, each iteration of the other algorithms are easy to parallelize.

**Densest Supermodular Set.** Efficient solvability of DSG can also be seen via a connection to a more general problem called the densest supermodular set problem, which we refer to as DSS. A real-valued set function $f : 2^V \to \mathbb{R}_+$ is said to be supermodular iff $f(A) + f(B) \leq f(A \cup B) + f(A \cap B)$ for all $A, B \subseteq V$. The goal in DSS is to return $S \subseteq V$ that maximizes $f(S)/|S|$. For any graph $G = (V, E)$, the function $f : 2^V \to \mathbb{R}_+$ defined as $f(S) = |E(S)|$ for each $S \subseteq V$, is known to be supermodular. Hence DSS generalizes DSG. Several algorithms and structural features for DSG are easier to understand via supermodularity, as shown in recent work [2].

*Notation:* For an undirected graph $G = (V, E)$ and $S_1, S_2 \subseteq V$, we use $E(S_1, S_2)$ for the edge set $\{\{u, v\} \in E : u \in S_1, v \in S_2\}$. For a vector $b \in \mathbb{R}^V$, and a subset $S \subseteq V$, we let $b(S)$ denote $\sum_{i \in S} b_i$. We let $ord(E)$ be the set of $2|E|$ edges that are ordered edges $uv$ and $vu$ for each unordered edge $\{u, v\} \in E$. Finally, $m$ and $n$ denote $|E|$ and $|V|$ respectively.

## 2. Technical Contributions

We summarize the main contributions of this paper. **A fast and scalable algorithm based on projections:** We describe an iterative algorithm for DSG and dense subgraph decomposition based on solving a quadratic objective with linear constraints that is derived from the dual of Charikar's LP. Unlike previous work [36] that relied on the Frank-Wolfe method, we use a projection-based approach. The algorithm is extremely simple and highly parallelizable. We show that it converges to an $\varepsilon$-additive approximation in $O(\sqrt{m\Delta(G)}/\varepsilon)$ iterations which is significantly better than the bound for the Frank-Wolfe method. The algorithm scales to extremely large graphs and outperforms existing algorithms in both running time and the approximation quality on almost all data sets. The algorithm and analysis generalizes to hypergraphs.

**Fractional Peeling**: We introduce the idea of fractional peeling to round a fractional solution to the underlying LP/QP relaxation and show its effectiveness in theory and practice. In experiments it significantly outperforms ordering based rounding algorithms considered previously. We use it to obtain a provably approximate dense subgraph decomposition from a fractional solution.

**Connections to DSS:** We explicitly connect the result of Fujishige [34] on the existence of a lexicographically optimal base in a polymatroid with the problem of computing a densest decomposition of a supermodular function. We show that the a vertex load vector is feasible for the dual of Charikar's LP for DSG iff it is feasible for the base contrapolymatroid associated with the supermodular func-

---

[4]Very recently the authors of this paper were able to prove the convergence of GREEDY++ to the dense decomposition. A proof will appear in a followup manuscript.

tion $f(S) = |E(S)|$ induced by the graph. These connections clarify past results such as the one in [36, 2] and allow us to compare Frank-Wolfe versus projection based methods.

**Experimental evaluation:** We compare five algorithms for DSG, namely GREEDY++, FRANK-WOLFE, MWU, FISTA, and FISTA-PARALLEL a parallel version of FISTA. We test the algorithms on real world and synthetic data sets and show the effectiveness of FISTA when combined with fractional peeling.

To make the paper self contained we provide all proofs in the appendix, even for results implicitly contained in past literature. We focus on DSG (unweighted), however, the ideas generalize to hypergraphs and weighted graphs, and we leave a more in-depth exploration for future work.

## 3. Densest Subgraph Decomposition and Supermodularity

Let $G = (V, E)$ be an undirected graph. Each graph admits a unique nested decomposition of decreasing densities. This property is more transparently seen via supermodularity. Let $f : 2^V \to \mathbb{R}_+$ be a non-negative supermodular set function[5]. For example, fix $f(S) = |E(S)|$ which is supermodular. Supermodularity implies that there is a unique inclusion-wise *maximal* densest set $S_1$. It is convenient to describe the decomposition in an algorithmic fashion. The algorithm calculates the maximal set $S_1 \subseteq V_0 = V$ that maximizes $\frac{|E(S)|}{|S|}$ in $G$ with density $\lambda_1 = \lambda^* = \frac{|E(S_1)|}{|S_1|}$. For DSS this corresponds to finding the unique maximal set $S_1$ that achieves the maximum density $\max_S f(S)/|S|$.

In iteration $i$, letting $U_{i-1} = \cup_{1 \leq j < i} S_j$, it calculates the maximal set $S_i \subseteq V_i = V - U_{i-1}$ that maximizes $(|E(S \cup U_{i-1}| - |E(\bar{U}_{i-1})|)/|S|$. The algorithm is described formally in Algorithm 1.

---

**Algorithm 1** Dense graph decomposition (left) and dense supermodular set decomposition (right)

---

| | |
|---|---|
| $U_0 \leftarrow \emptyset, V_0 \leftarrow V, k \leftarrow 0$ | $U_0 \leftarrow \emptyset, V_0 \leftarrow V, k \leftarrow 0$ |
| **while** $V_k \neq \emptyset$ **do** | **while** $V_k \neq \emptyset$ **do** |
| $\quad k \leftarrow k + 1$ | $\quad k \leftarrow k + 1$ |
| $\quad S_k \leftarrow \underset{\substack{S \subseteq V_{k-1} \\ S \text{ maximal}}}{\arg \max} \frac{|E(S)| + |E(S, U_{k-1})|}{|S|}$ | $\quad S_k \leftarrow \underset{\substack{S \subseteq V_{k-1} \\ S \text{ maximal}}}{\arg \max} \frac{f(S \cup U_{k-1}) - f(U_{k-1})}{|S|}$ |
| $\quad U_k \leftarrow U_{k-1} \cup S_k, V_k \leftarrow V_{k-1} - S_k$ | $\quad U_k \leftarrow U_{k-1} \cup S_k, V_k \leftarrow V_{k-1} - S_k$ |
| **return** $S_1, ..., S_k$ | **return** $S_1, ..., S_k$ |

---

The algorithm outputs a partition of $V$ into $S_1, ..., S_k$, where $k$ is the *dense decomposition depth* of $G$ (or any supermodular function $f$). With each set $S_i$, we associate a density defined as $\lambda_i = (|E(S_i)| + |E(S_i, U_{i-1})|)/|S_i|$ (or in the case of $f, \lambda_i = (f(S_i \cup U_{i-1}) - f(U_{i-1}))/|S_i|$). In addition, for any $u \in S_i$, we say the density of $u, \lambda_u = \lambda_i$. Refer to Lemma 6.2 (Appendix 6.1) for some basic properties of the dense decomposition. Specifically, the densities monotonically decrease, that is, $\lambda_1 > \lambda_2 > ... > \lambda_k$.

## 4. LP and QP for DSG and Decomposition

Charikar's [5] exact LP relaxation for DSG 4.1, and its dual 4.2 (in a slightly modified form), are given below. The primal has a variable $y_u$ for each vertex $u \in V$ which indicates whether $u$ is chosen in the densest set. For each edge $e = \{u, v\} \in E$ there is a variable $z_e$ to indicate whether it is chosen. An edge $\{u, v\}$ can be chosen only if both $u$ and $v$ are chosen which explains the constraints linking $z_e$ to $y_u, y_v$. The LP normalizes $\sum_u y_u$ to 1 for linearity, and maximizes chosen edges.

---

[5]We restrict attention to non-negative supermodular set functions that satisfy $f(\emptyset) = 0$ and this automatically also implies that they are monotone, that is, $f(A) \leq f(B)$ for $A \subset B$

$$\max \sum_{e \in E} z_e \qquad \text{(4.1)}$$

$$\text{s.t. } z_e \leq y_u \qquad \forall e = \{u, v\} \in E$$
$$z_e \leq y_v \qquad \forall e = \{u, v\} \in E$$
$$\sum_{u \in V} y_u \leq 1$$
$$z_e, y_u \geq 0$$

$$\min \max_{u \in V} b_u \qquad \text{(4.2)}$$

$$\text{s.t. } \sum_{v \in \delta(u)} x_{uv} = b_u \qquad \forall u \in G$$
$$x_{uv} + x_{vu} = 1 \qquad \forall \{u, v\} \in E$$
$$x_{uv}, x_{vu}, b_u \geq 0$$

The dual can be viewed as orienting each edge $\{u, v\}$ fractionally towards $u$ and $v$. The orientation induces loads on the vertices and the goal is to find an orientation that minimizes the maximum load over vertices. It is not hard to see that LP 4.2 is actually the same as the dual of 4.1; the variables $b_u$ can be replaced by a single variable $b$. The optimum value of LP 4.1 and LP 4.2 is the density $\lambda^* = \lambda_1$. However, we show that the $b_u$ values have additional information.

**Theorem 4.1.** *Let $S_1, ..., S_k$ and $\lambda_1, ..., \lambda_k$ be the densest subgraph decomposition of a graph $G$, and for any $u \in S_i$, let $\lambda_u = \lambda_i$. There is an optimal solution $(x^*, b^*)$ to 4.2 such that $b_u^* = \lambda_u$.*

We give two proofs, one a direct proof using Rado's theorem [37] in Appendix 6.1, and a second by relating 4.2 to another relaxation and using a known result of Fujishige that is captured in Theorem 4.3. We say a vector $a \in \mathbb{R}^n$ is *lexicographically* smaller than vector $b \in \mathbb{R}^n$ if the sorted vector $a$ (in descending order) is lexicographically smaller than the sorted vector $b$. Theorem 4.1 suggests that there exists a lexicographically least optimal solution to LP 4.2 where each vertex load $b_u$ is precisely $\lambda_u$. To obtain the lexicographic solution, it suffices to introduce some strict convexity into the objective. Let $P(x, b)$ denote the polyhedron defined by the constraints in 4.2. Consider the quadratic program 4.3:

$$\min \sum_{u \in V} b_u^2 \text{ such that } (x, b) \in P(x, b) \qquad \text{(4.3)}$$

The theorem below was shown by Danisch *et al.* [3] but as we will show later, this is a special case of Fujishige's result [34] from 1980 via Theorem 4.3.

**Theorem 4.2.** *There is a unique optimum solution $b^*$ to 4.3 and for each $u \in V(G), b_u^* = \lambda_u$.*

### 4.1. LP and QP for DSS and densest decomposition

We will now generalize Theorem 4.1 and Theorem 4.2 for the DSS problem. We start by recapping the notion of a contrapolymatroid (see [38]) which is the relevant notion for supermodular functions (as polymatroids are for submodular functions). For a normalized non-negative supermodular function $f : 2^V \to \mathbb{R}_+$, the contrapolymatroid with it is the following polyhedron

$$P_f = \{x \in \mathbb{R}^V \mid x \geq 0, \, x(S) \geq f(S) \text{ for all } S \subseteq V\} \qquad \text{(4.4)}$$

A vector $x \in P_f$ is a base if $x(V) = f(V)$. The base contrapolymatroid is defined as:

$$B_f = \{x \in \mathbb{R}^V \mid x \geq 0, \, x(S) \geq f(S) \text{ for all } S \subseteq V, \, x(V) = f(V)\} \qquad \text{(4.5)}$$

Now consider problem 4.6 where we are given a monotone non-negative supermodular function $f : 2^V \to \mathbb{R}_+$ and want to find the lexicographically minimal solution $b^*$ for Problem 4.6. We will show that we can do this by instead solving Problem 4.7:

$$\begin{aligned} \text{minimize} \quad & \max_{u \in V} b_u \\ \text{subject to} \quad & b \in B_f \end{aligned} \qquad \text{(4.6)}$$

$$\begin{aligned} \text{minimize} \quad & \sum_{u \in V} b_u^2 \\ \text{subject to} \quad & b \in B_f \end{aligned} \qquad \text{(4.7)}$$

**Theorem 4.3.** *Let $S_1, ..., S_k$ and $\lambda_1, ..., \lambda_k$ be the densest supermodular set decomposition of $f$, and for any $u \in S_i$, let $\lambda_u = \lambda_i$. Then the following must hold*

*1. The solution $b$ where $b_u = \lambda_u$ is feasible in the base polytope (i.e $b \in B_f$)*

2. *The lexicographically minimal solution $b^*$ for Problem 4.6 satisfies $b_u = \lambda_u$*
3. *The optimal solution of 4.7, $b^*$ is unique, and for each $u \in V, b_u^* = \lambda_u$*

We give a self-contained proof in Appendix 6.1 and note that the theorem is implied by (essentially equivalent to) Fujishige's result [34] on the existence of a lexicographically optimal base of a polymatroid with respect to a weight vector. LP 4.2 and LP 4.6 can be related via the following theorem.

**Theorem 4.4.** *Consider a graph $G = (V, E)$ and the associated supermodular function $f : 2^V \to \mathbb{R}_+$ where $f(S) = |E(S)|$. A vector $b \in B_f$ if and only if there is an $x \in \mathbb{R}^{ord(E)}, x \geq 0$ such that the pair $(x, b)$ satisfy the constraints of the LP 4.2.*

See Appendix 6.1 for a proof. This implies that Theorem 4.2 is a corollary of Theorem 4.3.

## 5. Solving the Quadratic Program using proximal projections, and rounding

In this section we show how to approximately solve 4.3. We let $f(x) = \sum_{u \in V} \left( \sum_{v \in \delta(u)} x_{uv} \right)^2$. Note that this is simply the objective function rewritten in terms of $x$. Similarly, let $h(x)$ be an indicator function where $h(x) = 0$ if $x_{uv} \geq 0$ and $x_{uv} + x_{vu} = 1, \forall(u, v) \in E$ and $+\infty$ otherwise. Then Problem 4.3 can be rewritten as minimizing the unconstrained objective $f(x) + h(x)$ for $x \in \mathbb{R}^{2m}$. We will use a proximal gradient method to solve the problem. For that, we need two lemmas whose proofs are in Appendix 6.2.

**Lemma 5.1.** *The Lipschitz constant of $\nabla f$ is at most $2\Delta(G)$ where $\Delta(G)$ is the max degree of $G$.*

**Lemma 5.2.** *Let $x \in \mathbb{R}^{2m}$. Define the proximal mapping $prox_h(x)$ as the point $p \in \mathbb{R}^{2m}$ that minimizes $\|p - x\|^2$ such that $h(p) = 0$. Then we have that for $u < v$, $prox_h(x)_{uv} = \frac{x_{uv} - x_{vu} + 1}{2}$ if $|x_{uv} - x_{vu}| \leq 1$, $prox_h(x)_{uv} = 1$ if $x_{uv} - x_{vu} > 1$, and $prox_h(x)_{uv} = 0$ if $x_{uv} - x_{vu} < -1$. Additionally, $prox_h(x)_{vu} = 1 - prox_h(x)_{uv}$.*

We present the algorithm now. We are interested in the unconstrained optimization problem of minimizing $f(x) + h(x)$ where $f$ is convex, and $h$ has an easy to compute proximal mapping. This type of problem can be solved using proximal gradient methods. From a high level, the (basic) algorithm is described Algorithm 2. At any iteration $t$ it has a guess for the minimizer $x^{(t)}$. It then calculates the gradient of $f$ and moves slightly against it. However, since this might make the new guess infeasible, it uses the proximal mapping to project the new guess to a feasible solution.

---

**Algorithm 2** Basic Proximal Gradient Method (left) and accelerated FISTA (right)

---

Input: $f$ and $h$ with $prox_h(x)$, learning rate $\alpha$ and iterations $T$. Initialize $x^{(0)}$ with $h(x^{(0)}) = 0$

$\quad$**for** $t \in [1, T]$ **do** $\qquad\qquad\qquad\qquad\quad$ $y^{(0)} = x^{(0)}$
$\qquad x^{(t)} = prox_h(x^{(t-1)} - \alpha \nabla f(x^{(t-1)}))$ $\qquad$**for** $t \in [1, T]$ **do**
$\quad$**return** $x^{(T)}$ $\qquad\qquad\qquad\qquad\qquad\qquad\quad x^{(t)} = prox_h(y^{(t-1)} - \alpha \nabla f(y^{(k-1)}))$
$\qquad\qquad\qquad\qquad\qquad\qquad\qquad\qquad\qquad y^{(t)} = x^{(t)} + \frac{t-1}{t+2}(x^{(t)} - x^{(t-1)})$
$\qquad\qquad\qquad\qquad\qquad\qquad\qquad$**return** $x^{(T)}$

---

While the basic proximal gradient method works, we will use an even faster (both theoretically and practically) version known as the accelerated proximal gradient method which incorporates Nesterov-like momentum terms [39] in the projection step. It has other names in the literature such as proximal gradient method with extrapolation and FISTA [40]. The algorithm is outlined in Algorithm 2. We have the following known result on the FISTA algorithm.

**Lemma 5.3.** *[40]. Let $x^*$ be the minimizer of $f$. Suppose that the learning rate satisfies $\alpha \leq \frac{1}{L(f)}$ where $L(f)$ is the Lipschitz constant of $\nabla f$. Then after $k$ iterations, $f(x^{(k)}) - f(x^*) \leq \frac{2\|x^{(0)} - x^*\|^2}{\alpha k^2}$.*

In our case, we bounded the Lipschitz constant to $2\Delta(G)$ and an easy upper bound on $\|x^{(0)} - x^*\|^2$ is $2m = 2|E(G)|$. Combining this with a learning rate of $\frac{1}{2\Delta(G)}$, we get the following result

**Lemma 5.4.** *If FISTA is applied to the objective $f(x) + h(x)$ as previously defined, then in the kth iteration, we have $f(x^{(k)}) - f(x^*) \leq \frac{8\Delta(G)m}{k^2}$.*

The final FISTA algorithm for our problem is shown in Algorithm 3 in Appendix 6.2 with the correct gradient computation, projective mappings, and iteration updates. Lastly, to get a good additive approximation on each $\lambda_u$, we have the following result whose proof is in Appendix 6.3

**Theorem 5.5.** *Let $b_u^* = \lambda_u$. After $t = O(\frac{\sqrt{\Delta(G)m}}{\epsilon})$ iterations of FISTA, we must have $\left\| b^{(t)} - b^* \right\| \leq \epsilon$. This implies $\left| b_u^{(t)} - \lambda_u \right| \leq \epsilon$ for all $u \in V$*

## 5.1. Fractional Peeling.

Given an approximate solution $(b, x)$ to Problem 4.3, how do we round it to obtain a good densest decomposition? The natural approach is to sort the vertices in non-increasing values of $b$ and take suffixes. This is used in [36]. This is an exact rounding algorithm when $b$ is an *optimum* solution, however one can construct approximate solutions for which this rounding is not ideal. We describe a peeling algorithm inspired by GREEDY++ [1] that takes advantage of the auxiliary information provided by the vector $x$. First, set $b' = b$ and $G^{(0)} = G$. In iteration, $t \geq 1$, peel the vertex $u$ with minimum current load $b_u'$. Then, for each $v \in \delta_{G^{t-1}}(u)$, set $b_v' \leftarrow b_v' - x_{vu}$ (i.e subtract the fractional value of $x_{vu}$ from $v$'s load). Update $G^t = G^{t-1} - u$. Repeat this process to obtain $G^{(0)}, ..., G^{(n)}$. Finally, return a graph $G^{(i)}$ with maximum density. The algorithm can be implemented in $O(m + n \log n)$ using a Fibonacci heap. We will refer to this process as *fractional peeling*. We show both theoretically and experimentally that fractional peeling leads to better algorithms.

## 5.2. $\epsilon$-dense local decomposition

We show that fractional peeling can be used on an approximate solution to 4.3 to obtain an approximate dense decomposition with a theoretical guarantee. This is in contrast to previous work in [36]. We define a strong notion of approximate decomposition. Given a solution $(b, x)$ to 4.3, we say $\hat{b}$ is an $\epsilon$-*load-vector* if $\left\| \hat{b} - b^* \right\| \leq \epsilon$. Given a partition of the vertices $T_1, ..., T_r$, we say the partition is an $\epsilon$-*approximate dense decomposition* to $S_1, ..., S_k$ (the true dense decomposition) if

$$u \in S_i, u \in T_h \implies \frac{|E(T_h)| + |E(T_h, \cup_{j<h}T_j)|}{|T_h|} \geq \frac{|E(S_i)| + |E(S_i, \cup_{j<i}S_j)|}{|S_i|} - \epsilon$$

Intuitively, what this says is that every $u \in V$ belongs to a set $T_h$ that has a density that is not much less than $\lambda_u$.

**Theorem 5.6.** *Given an $\epsilon$ load vector $b$ and an edge vector $x$ that induces $b$, we can calculate an $\epsilon(\sqrt{n} + 1)$-approximate dense decomposition in $\tilde{O}(mn)$ time.*

The proof is in Appendix 6.4. Note that the notion of error is additive and holds for every vertex $u$. Although the $\sqrt{n}$-factor is large, the analysis shows that one can usually obtain a much stronger bound. Qualitatively it shows that fractional peeling leads to good dense decomposition as $\varepsilon \to 0$.

## 5.3. Projections vs Frank-Wolfe vs MWU for DSG and DSS

The Frank-Wolfe method is natural to apply to solve 4.7 since each iteration requires optimizing a linear objective over the base contrapolymatroid $B_f$; this is easy and fast via the greedy algorithm, as shown originally by Edmonds in the context of polymatroids (see [38]). Danisch *et al.* work with 4.3. However, as we observed earlier, 4.3 is a compact way to represent the associated base polyhedron, and hence the algorithm in [36] is the same as the Frank-Wolfe algorithm applied to 4.7. The Fujishige-Wolfe minimum norm point algorithm for submodular function minimization is related but is based on Wolfe's method (see [41, 42]). The fact that the optimum point in the base polytope has additional information was already pointed out in [34], and also explored in the context of size-constrained submodular function minimization by Nagano *et al.* [33]. The advantage of proximal gradient methods such as FISTA is their faster convergence rates when compared to the Frank-Wolfe method, although each iteration requires a projection oracle for $B_f$. Our algorithm is

based on the observation that there is an $O(m)$-time projection oracle for DSG due to the alternate characterization of $B_f$ via the edge variable LP 4.2. This same observation holds for hypergraphs and we obtain a fast algorithm for them. We outline the formal details in Appendix 6.5. The multiplicative weight update (MWU) method is another broad methodology for solving linear and convex programs and there are several variants. One variant of MWU for solving 4.2 can be interpreted as a Frank-Wolfe method with a certain convex objective and with a certain step size; each iteration of this algorithm is again a greedy optimization over $B_f$. We provide more details in the Appendix 6.7.

### 5.4. A theoretically fast algorithm for approximate load vector via min-cost flow

While Algorithm 3 (detailed in Appendix 6.2) works extremely well in practice, there is room to improve the theoretical run time. Each iteration takes $O(m)$ time and can easily be parallelized. However, in the worst-case it would need $O(m\sqrt{\Delta(G)m}/\epsilon)$ time to get an additive $\epsilon$-approximation. Can we obtain a theoretically faster algorithm? In Appendix 6.6 we describe a reduction to computing *exact* minimum quadratic-cost flow in a directed network. Very recently, in a breakthrough, [43] developed near-linear time algorithm for min-cost flow and also convex-cost flows. Via their algorithm we obtain an $O(m^{1+o(1)})$ time algorithm to get an optimal dense decomposition vector. We hope that this theoretical result will inspire the development of new algorithms that are provably faster in theory while also having good empirical performance.

### 5.5. Frank-Wolfe and MWU

The Frank-Wolfe algorithms is part of the broader family of algorithms referred to as conditional linear gradient methods. These algorithms approximately minimize a convex function $f(x)$ over a polytope $P$ on which one can do efficient linear optimization. Generally speaking, each iteration $t$, these algorithms start with the current point $x_{t-1} \in P$ and solve the linear optimization problem $\min_{y \in P}(\nabla f(x_{t-1})^T y$. Let $p_t$ be optimum solution to this; the algorithm sets $x_t$ to be a convex combination of the current point and this vertex, $x_t = \alpha_t p_t + (1 - \alpha_t)x_{t-1}$. The parameter $\alpha_t$, called the step size, controls the convergence rate. The Frank-Wolfe method often refers to the specific step size of $\alpha_t = 2/(t + 2)$ [44]. Other possibilities including $\alpha_t = 1/t$, which implies that each $x_t$ will be a uniform combination of all previous vertices $p_1, p_2, \ldots, p_t$. This gives a slower rate of convergence proportional to $\ln t/t$, rather than $1/t$ (see for instance [45]).

An alternative continuous approach, popular for solving obtaining multiplicative approximations to LP's, is the *multiplicative weight update* framework [46]. There are several ways to leverage the MWU framework for the densest subgraph problem (e.g., [30, 1, 2]). We test one variation that applies the MWU framework to solve LP 4.2 that minimizes the maximum load. In this case, the MWU framework implicitly tries to minimize a potential function that exponentiates these loads and sums them together. More formally, for a parameter $\eta > 0$, consider the problem $\min \frac{1}{\eta} \ln(\sum_{u \in V} \exp(\eta b_u))$ over $b \in B_f$. As $\eta \to \infty$ one can see that the optimum solution to this problem converges to the minimum load vector $b^*$. Each iteration of the MWU framework involves solving a simpler linear optimization problem induced by the gradient of this potential. In the specific context of DSS the linear optimization problem corresponds to the greedy algorithm over $B_f$. The greedy algorithm only depends on the ordering of $V$ based on the current loads (and not their specific values). For this reason, when the MWU method is applied with a fixed step size (in the so-called "width-dependent MWU framework"), the MWU algorithm ends up solving the exact same sequence of optimization problems, in the exact same way, as the conditional linear gradient method with step size $\alpha_t = 1/t$ would for the objective $\sum_u b_u^2$! Thus the parameter $\eta$ does not a play a role in this specific case, and it also follows that the MWU algorithm converges to an $\epsilon$-approximate load vector, although at a slightly lower rate than the Frank-Wolfe method.

## 6. Experimental Evaluation

**Datasets & Implementation Details.** We ran experiments on 7 real world datasets (6 from the SNAP database [47], and 1 from [48]), and one tailored synthetic dataset (used for clarifying an important difference between all algorithms) for a total of 8 datasets. The dataset information is summarized in the table below. The CLOSE-CLIQUES dataset consists of the complete bipartite graph $K_{d,D}$ for $d = 30$ and $D = 2000$, and 20 copies of the complete graph $K_h$ where $h = 60$.

| Dataset | Vertices | Edges | Source |
|---|---|---|---|
| cit-Patents | 3,774,768 | 16,518,947 | [47] |
| com-Amazon | 334,863 | 925,872 | [47] |
| orkut | 3,072,441 | 117,185,083 | [48] |
| roadNet-PA | 1,088,092 | 1,541,898 | [47] |

| Dataset | Vertices | Edges | Source |
|---|---|---|---|
| roadNet-CA | 1,965,206 | 2,766,607 | [47] |
| Close-Cliques | 3,230 | 95,400 | Synthetic |
| dblp-author | 317,080 | 1,049,866 | [47] |
| wiki-topcats | 1,791,489 | 28,508,141 | [47] |

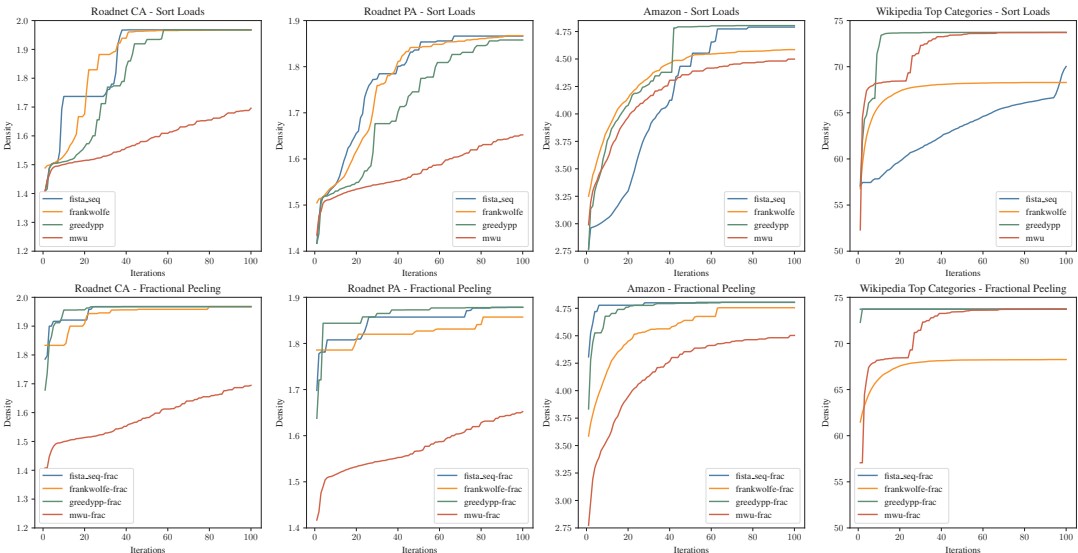

Figure 6.1: Density based on sorting loads vs fractional peeling.

We tested 5 algorithms to approximate DSG: The FISTA based algorithm (sequential, no parallelism), FISTA-PARALLEL, GREEDY++ from [1], the FRANK-WOLFE based algorithm from [3], and the MWU algorithm respectively. See supplementary section for implementation details of the 5 algorithms. Also see Appendix 6.7 for details on the specific variant of MWU we tested. All algorithms were implemented in C++17 and were compiled with O3 and UNROLL-LOOPS optimizations. The implementations of Algorithms FISTA, FISTA-PARALLEL, FRANK-WOLFE, and MWU are the authors' implementations, but we used the original implementation for GREEDY++ [1] as it was extremely well optimized. We modified their implementation minimally to log basic information needed for the evaluation. FISTA-PARALLEL used Open MPI [49] for parallelism. We ran our experiments on a Slurm-based university campus cluster. For all machines, we requested 1 node and 16 cores per experiment. The nodes had 64 GB of RAM and Xeon PHI 5100 CPUs.

**Densities**. In the first experiment, we ran all the algorithms for a number of iterations, and monitored the maximum density reached by the algorithm until iteration $t$. The density in each iteration was calculated by statically sorting the vertices by the value of their load vector, and peeling in that order and returning the maximum density subgraph in that iteration. The result is shown in Figure 6.1 (Top 4 plots, we only show 4 datasets, remaining plots in Appendix 6.7). Specifically, this does not use the fractional peeling technique we discussed. We also plot the maximum density reached by using fractional peeling instead (Bottom 4 plots in 6.1, we show 4 datasets, remaining plots in Appendix 6.7).

When the peeling order is changed from a static peel (based on the load vector value) to a fractional peel, the results improve dramatically. See Figure 6.1 and Appendix 6.7. In all algorithms (except MWU), fractional peeling leads to a dramatic speedup in terms of the number of iterations needed to get the densest subgraph on almost all datasets. Fractional peeling provides little benefit to MWU. We can observe that for all the real world datasets, GREEDY++ and FISTA are extremely competitive and reach near-optimal densities in just a few iterations. Meanwhile, the FRANK-WOLFE based algorithm lagged behind in the beginning, but steadily made progress towards the maximum density in later iterations. MWU strongly lagged behind all algorithms in several datasets. The only excep-

tion is for the CLOSE-CLIQUES dataset (see Appendix 6.7). The component in the densest subgraph has density $\approx 29.5566$. Meanwhile, the remaining components have density $\binom{60}{2}/60 = 29.5$. This is a similar but nonidentical to the example from [1]. Note that for $K_{d,D}$ and $r$ copies of $K_{2d}$ (for sufficiently large $r, D$), $\exists\epsilon$ where GREEDY++ requires $\Theta(\frac{1}{\epsilon})$ iterations to converge to a $(1-\epsilon)$ approximation for DSG in the worst case. MWU and FRANK-WOLFE did better for this synthetic example where the densities are very close and it is worth understanding in more detail.

**Wall clock time and run-time per iteration.** In the second experiment we examine the wall-clock time and time per iteration. We study the maximum density reached by an algorithm after $T$ seconds of wall-clock time (cumulative time of all iterations). Figure 6.13 (Appendix 6.7) and last 4 plots of Figure 6.2 shows the result. As can be seen, FISTA-PARALLEL and FISTA (sequential) finds the maximum density on almost all datasets in the least wall-clock time (albeit sequential FISTA and FRANK-WOLFE are close runners if we only restrict to non-parallel implementations). On the other hand, MWU performed poorly in wall-clock time, often taking orders of magnitude longer than FISTA or GREEDY++ to find suboptimal dense subgraphs. Overall, the number of iterations of FISTA is the lowest, and each iteration is fast and can be parallelized, so it is the best performer.

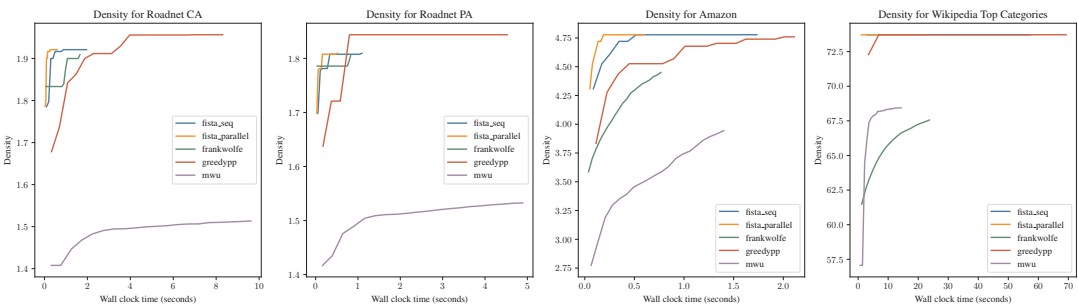

Figure 6.2: Wall clock time of algorithms.

We also considered the run time per iteration for the algorithms. Figure 6.9 (appendix) shows the result. The per iteration time depends on several factors. Although all algorithms have linear or near-linear in $m$ running time, the specific data structure used and cache performance can have substantial impact. Overall, FRANK-WOLFE (due to its simple implementation) and FISTA-PARALLEL had the best per-iteration performance. The average speedup *per iteration* from FISTA-PARALLEL over GREEDY++ was roughly 5 fold. Note that due to the peeling nature of GREEDY++, it cannot be parallelized like FISTA.

**Convergence to optimal load vector.** Recall that we are minimizing $\sum_u b_u^2$, and hence the norm $\|b^{(t)}\|$ gives a proxy for the convergence of the errors in $b^{(t)}$. In this experiment, we plotted the norm of the vector $b^{(t)}$ for each algorithm in each dataset. The result is shown in Figure 6.10 (Appendix 6.7). GREEDY++ does very well in the first few iterations, but slows down in its improvement on $\|b^{(t)}\|$ as the number of iterations increases. In comparison, FISTA eventually surpasses GREEDY++ and reduces the error at a faster rate than GREEDY++. In comparison, FRANK-WOLFE and MWU start with a substantial error in the $b$ vector but quickly reduce it, however, their error was always worse that both GREEDY++ and FISTA in all iterations even when left to run 200 iterations.

**Conclusion.** We introduced a new iterative algorithm for the densest subgraph and densest decomposition problems. We also described a new fractional peeling technique which has strong empirical performance as well as theoretical guarantees and showed experimentally how it improved almost all existing algorithms compared to static load sorting. The new algorithm is scalable and simple, and can be applied to graphs with hundreds of millions of edges. Our experiments support the theory established on the utility of the new algorithm and fractional peeling. Our work also adds value via a detailed comparison of the practical performance of existing algorithms and the new algorithm. A few limitations remain for the paper. First, reducing the $\sqrt{n}$ dependency in Theorem 5.6 is important, even if fractional peeling shows strong experimental bounds. Finally, we need to evaluate empirically the quality of the decomposition achieved by algorithms (with and without fractional peeling), and also algorithms for approximating at least $k$ DSG and related problems via the decomposition.

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
