# OpenReview forum: "Faster and Scalable Algorithms for Densest Subgraph and Decomposition"
_NeurIPS.cc/2022/Conference — NeurIPS 2022 Accept_

### Official Review · Reviewer_WcJR · 2022-07-10

**Rating:** 7
**Confidence:** 4
**Soundness:** 3 good
**Presentation:** 3 good
**Contribution:** 3 good

**Summary:**

The authors present a new algorithm for densest subgraph (DSG) and graph decomposition  (DSG-LD) in undirected graphs by leveraging (1) proximal method and (2) fractional peeling inspired by recent work of GREEDY++ which uses information from load vector of the dual LP for a modified iterative peeling. The method is shown theoretically and empirically to be faster than competing methods and the fractional peeling improves quality of the solution compared to direct ordering.

**Questions:**

- Lines 309-311 is not new information and may/should be removed? Poor performance of Greedy++ for CloseCliques is a known example which was highlighted in the original [Boob+] paper and later expanded upon in the [Chekuri+, SODA '22] paper.


**Limitations:**

Yes, discussed in Conclusion L347-351

**Strengths And Weaknesses:**

+ Well-explained paper starting from Charikar's dual LP with clear description of connection to supermodular functions and results therein. Parts of this are known (e.g. Danisch et al and Chekuri et al, SODA 2022) but the relevant parts for this paper are clearly presented.
+ Paper adapts iterative peeling based on dual load vector idea introduced in Boob et al. to the DSG-LD problem.
+ Adapts proximal method for DSG and DSG-LD with theoretical results
+ Detailed empirical evaluation with good comparison highlighting different scenarios.
+ (Minor) Theoretical connection to the breakthrough result of [Peng et al, 2022] which promises new avenues for development of practical algorithms.
+ (Minor) Extension of hypergraphs discussed in appendix 6.7

The discussion in the main text is good but the proofs could be better illustrated in places (See some suggestions below).

- Section 6.1 (Rado, b-transportation) Suggest adding a figure showing B(V', E') for easier comprehension.
- Section 6.5 - please refer explicitly from main text in Section 5.4 (lines 266 or 269). Also the order of the appendix section and main text is inverted so suggest changing order of 6.5 and 6.7
- Line 260. "outline the formal details in appendix `6.7`"
- Section 6.4 - Proof of 5.6 requires very careful reading. Suggest adding a figure/Venn diagram showing T_t and summation in 619.

---

> ### Author Response · Authors · 2022-08-01
> **Reply to Reviewer WcJR**
>
> > Section 6.1 (Rado, b-transportation) Suggest adding a figure showing B(V', E') for easier comprehension.
>
> Thank you for the recommendation. We added a figure (Figure 6.3 in the new Appendix) to clarify the proof.
>
> > Section 6.5 - please refer explicitly from main text in Section 5.4 (lines 266 or 269). Also the order of the appendix section and main text is inverted so suggest changing order of 6.5 and 6.7
>
> We reordered sections in the Appendix as per the suggestion, and we now explicitly reference the relevant Appendix section in the main body of the paper.
>
> > Line 260. "outline the formal details in appendix 6.7
>
> This is now fixed throughout the paper. Any part in the main text that references the Appendix has the relevant Appendix section number.
>
> > Section 6.4 - Proof of 5.6 requires very careful reading. Suggest adding a figure/Venn diagram showing $T_t$ and summation in 619.
>
> Thank you for the recommendation. We have added Figure 6.4 in the Appendix to help clarify the proof. We note that this proof is a cleaner rewrite of the original version we had, and while a figure will help, the proof remains subtle and requires, as mentioned a careful read.
>
> > Lines 309-311 is not new information and may/should be removed? Poor performance of Greedy++ for CloseCliques is a known example which was highlighted in the original [Boob+] paper and later expanded upon in the [Chekuri+, SODA '22] paper
>
> The lines were rewritten to clarify the meaning. In the original Boob et al. paper, the example used is $K_{d,D}$ and $k$ copies (for large $k$) of $K_{d+2}$ (note the $d+2$) which illustrates why Greedy++is a significant **improvement** over the classical greedy algorithm (indeed it takes only $2$ iterations to get a very good approximation). Our example is slightly different ($K_{d,D}$ but $k$ copies (for large $k$) of $K_{2d}$)). This example is to show a provably worst case example for Greedy++ (which was not clear to the authors prior to writing this paper). This simply states that there is a class of graphs $G(d, D, r)$ such that $\exists \epsilon = O(\frac{d}{D})$ where Greedy++ provably requires $\Theta(\frac{D}{d})=\Theta(\frac{1}{\epsilon})$ iterations to reach a $(1-\epsilon)$-approximation to DSG. In fact, we know of no provable examples showing that Greedy++ requires $\omega(\frac{1}{\epsilon})$ iteration to reach a $(1-\epsilon)$ approximation to DSG. The example shows that Greedy++ ``true'' approximation ratio, after $T$ iterations, is between $O(1+\frac{1}{T})$ and $O(1+\frac{\Delta(G)}{\lambda^\ast \sqrt{T}})$ (the upper bound proven by Chekuri et al.). The Chekuri et al. paper also does not discuss lower bounds on Greedy++.
>
> If we have misunderstood the reviewer's intention  or missed details from previous papers, we are happy to revise and edit these lines. This is not directly relevant to the contributions of the paper, and is mainly of interest in a full theoretical understanding of Greedy++ (which is a fascinating question to us).

---

### Official Review · Reviewer_BbXa · 2022-07-11

**Rating:** 7
**Confidence:** 2
**Soundness:** 3 good
**Presentation:** 3 good
**Contribution:** 3 good

**Summary:**

This paper considered the densest subgraph, densest subgraph decomposition, and supermodular generalizations of these problems. They provided a fast and scalable algorithm that can compute a $\epsilon$ approximate local decomposition vector in $O(\sqrt{m\Delta(G)}/\epsilon)$ iterations, where $\Delta(G)$ is the maximum degree of graph G.
They also showed that a fractional peeling algorithm can round the fractional solution to get an $\epsilon\sqrt{n}$ approximation dense decomposition.

**Questions:**

1. Is the improvement in the number of iterations to convergence mainly coming from the connection between the LP dual and the QP shown in Thm 4.3?

**Limitations:**

Yes.

**Strengths And Weaknesses:**

Strengths:
The densest subgraph and densest subgraph decomposition are classic and important problems. This paper provided a fast and scalable algorithm that reduces the number of iterations to converge to an $\epsilon$ approximate local decomposition vector. They showed an interesting connection between LP and its dual and QP.
The paper is well-written and easy to follow.

---

> ### Author Response · Authors · 2022-08-01
> **Reply to Reviewer BbXa**
>
> > Is the improvement in the number of iterations to convergence mainly coming from the connection between the LP dual and the QP shown in Thm 4.3?
>
> The *theoretical* improvement in the number of iterations for convergence comes from two sources. The first, as you mention, is the characterisation in Thm 4.3. The  other is the fact that there is an **efficient** closed form proximal projection mapping (using Thm 4.3) which is reported in Lemma 5.2 for the base contra-polymatroid corresponding to DSG. The projection oracle is easy for the specific case of DSG (and more generally hypergraphs) while it is not necessarily easy for general supermodular functions (even though the QP formulation still works for any supermodular function). Both lead to a theoretically faster algorithm. The *practical* improvement is driven by Thm 4.3 and the fractional peeling technique we introduced (Section 5.1). We give a theoretical justification for the goodness of fractional peeling in Thm 5.6.

---

> > ### Comment · Reviewer_BbXa · 2022-08-09
> > **To Authors**
> >
> > Thanks for the response. My question is well addressed.

---

### Official Review · Reviewer_ZguU · 2022-07-11

**Rating:** 7
**Confidence:** 3
**Soundness:** 3 good
**Presentation:** 3 good
**Contribution:** 3 good

**Summary:**

The authors of this paper propose an iterative algorithm for densest subgraph detection and decomposition, where the proposed algorithm is guaranteed to converge to an \epsilon-additive approximate solution and the number of iterations needed is significantly smaller than [3]. Besides the theoretical contributions, the authors conduct extensive experiments on several large-scale real-world graphs to show the practical effectiveness and efficiency of their proposed algorithm.

**Questions:**

I do not have questions so far.

**Limitations:**

1. It would be better if the authors could list the approximation guarantee and the time complexity of each existing algorithm in a Table. Then readers can easily compare the theoretical guarantees of this paper and existing studies.
2. Although optimization definitely is an important topic in NeurIPS, my understanding is that optimization papers published in NeurIPS majorly focus on optimization problems closely related to machine learning. It seems to me that dense subgraph detection is not directly related to major machine learning problems and it is more extensively studied in the data mining community or the algorithm design community. The weak relevance of this paper to NeurIPS is the major reason that I give a score of 6 for this paper.

**Strengths And Weaknesses:**

1. The proposed algorithm significantly improves the algorithm in [3].
2. The results of this paper also apply to the densest decomposition of a supermodular function, which is a more general problem.
3. The authors also conduct extensive experiments on a number of large-scale graphs. The experimental results demonstrate that the proposed algorithm is not only theoretically sound but also highly effective and efficient in practice.

---

> ### Author Response · Authors · 2022-08-01
> **Reply to Reviewer ZguU**
>
> > It would be better if the authors could list the approximation guarantee and the time complexity of each existing algorithm in a Table. Then readers can easily compare the theoretical guarantees of this paper and existing studies.
>
> Thank you for the recommendation. Please see Figure 6.4 in the new Appendix (Section 6.1). We have also referenced the summary table in the Introduction of the main paper. We plan to describe the summary table in the main body  of the final version of the paper --- this is difficult to do now due to the 9 page limit.
>
> > Although optimization definitely is an important topic in NeurIPS, my understanding is that optimization papers published in NeurIPS majorly focus on optimization problems closely related to machine learning. It seems to me that dense subgraph detection is not directly related to major machine learning problems and it is more extensively studied in the data mining community or the algorithm design community. The weak relevance of this paper to NeurIPS is the major reason that I give a score of 6 for this paper.
>
> We thank the reviewer for clarifying the rationale behind their score. We explain why we see NeurIPS as a suitable venue for our paper.
>
> 1) Dense subgraph discovery is an important topic that is studied in several areas of CS including machine learning.  We point out below
>         several papers on densest subgraph,  $k-$core decomposition (very closely related to dense subgraph decomposition in this paper), and dense subgraph discovery (citations [1-11]) that have been recently published at ML conferences (NeurIPS/ICML/ECML PKDD/...). The topic is also closely related to graph clustering and the techniques are of interest to several sub-areas in ML and data analysis.
> 2) While traditional algorithms for DSG and DSG-LD were combinatorial in nature or relied on black-box reduction to max-flow, more recent work, including ours, has shown the importance of continuous optimization techniques and iterative methods. Ours is the first to use an accelerated gradient descent based algorithm for DSG-LD. Second, our paper is also closely connected to sub/supermodular optimization which are topics of much interest in ML. We believe that the paper is appealing to several researchers in ML and Optimization at the technical level. For example, the NeurIPS 2019 paper [12] on faster algorithms for positive LPs was essentially motivated by the application to DSG. As another example, while writing the paper, in private correspondence, an ML researcher focusing on optimization in deep learning expressed interest in the problem from a pure optimization point of view. In fact, some of the experimental modifications were guided by ideas shared during the correspondence. (The researcher is acknowledged in the deanonymized version of this paper). There are several unanswered questions that have the potential to capture the attention of ML optimization researchers.
>
>     We hope that you will reconsider the score in light of our explanation.

---

> > ### Author Response · Authors · 2022-08-01
> > **Citation list in reply to Reviewer ZguU**
> >
> >         [1]:  Kuroki, Y., Miyauchi, A., Honda, J. Sugiyama, M. (2020). Online Dense Subgraph Discovery via Blurred-Graph Feedback. Proceedings of the 37th International Conference on Machine Learning, in Proceedings of Machine Learning Research.
> >
> >         [2]: Nguyen, D. Vullikanti, A. (2021). Differentially Private Densest Subgraph Detection. Proceedings of the 38th International Conference on Machine Learning, in Proceedings of Machine Learning Research.
> >
> >         [3]: Jethava, Vinay and Martinsson, Anders and Bhattacharyya, Chiranjib and Dubhashi, Devdatt. (2012). The Lov\'{a}sz $\theta$ function, SVMs and finding large dense subgraphs. Advances in Neural Information Processing Systems. (NIPS 2012).
> >
> >         [4]: Alsentzer, Emily and Finlayson, Samuel and Li, Michelle and Zitnik, Marinka. Subgraph Neural Networks (2020). Advances in Neural Information Processing Systems.
> >
> >         [5]: Papailiopoulos, Dimitris S. and Mitliagkas, Ioannis and Dimakis, Alexandros G. and Caramanis, Constantine (2014). Finding Dense Subgraphs via Low-Rank Bilinear Optimization. Proceedings of the 31st International Conference on Machine Learning - Volume 32.
> >
> >         [6]: Jethava, V., Beerenwinkel, N. (2015). Finding Dense Subgraphs in Relational Graphs. In: Appice, A., Rodrigues, P., Santos Costa, V., Gama, J., Jorge, A., Soares, C. (eds) Machine Learning and Knowledge Discovery in Databases. ECML PKDD 2015
> >
> >         [7]: Ghaffari, M., Lattanzi, S., Mitrovic, S. (2019). Improved Parallel Algorithms for Density-Based Network Clustering. ICML.
> >
> >         [8]: Feng, W., Liu, S., Koutra, D., Shen, H., Cheng, X. (2021). SPECGREEDY: Unified Dense Subgraph Detection. In: Hutter, F., Kersting, K., Lijffijt, J., Valera, I. (eds) Machine Learning and Knowledge Discovery in Databases. ECML PKDD 2020
> >
> >         [9]: J. Ignacio Alvarez-Hamelin, Luca Dall'Asta, Alain Barrat, and Alessandro Vespignani. 2005. Large scale networks fingerprinting and visualization using the k-core decomposition. In Proceedings of the 18th International Conference on Neural Information Processing Systems (NIPS'05). MIT Press, Cambridge, MA, USA, 41–50.
> >
> >         [10]: Rozenshtein, P., Tatti, N., Gionis, A. (2014). Discovering Dynamic Communities in Interaction Networks. In: Calders, T., Esposito, F., Hüllermeier, E., Meo, R. (eds) Machine Learning and Knowledge Discovery in Databases. ECML PKDD 2014
> >
> >         [11]: H. Esfandiari, S. Lattanzi and V. Mirrokni. Parallel and Streaming Algorithms for K-Core Decomposition. ICML 2018.
> >
> >         [12]: Digvijay Boob, Saurabh Sawlani, and Di Wang. 2019. Faster width-dependent algorithm for mixed packing and covering LPs. Proceedings of the 33rd International Conference on Neural Information Processing Systems.

---

> > > ### Comment · Reviewer_ZguU · 2022-08-07
> > > **Response to the authors**
> > >
> > > Thanks for the clarification. I would be happy to revise my score to 7.

---

> > > > ### Author Response · Authors · 2022-08-07
> > > > **Repsponse to Reviewer ZguU**
> > > >
> > > > We thank the reviewer for taking the time to read the clarification and reconsidering their score.

---

### Official Review · Reviewer_wCmG · 2022-07-12

**Rating:** 7
**Confidence:** 2
**Soundness:** 3 good
**Presentation:** 3 good
**Contribution:** 3 good

**Summary:**

The authors present a new iterative algorithm for Densest subgraph (DSG) and Densest Subgraph Decomposition (DSG-LD) problems. The algorithm takes $O(\frac{\sqrt(m \Delta(G))}{\epsilon})$ iteration to converge to an $\epsilon$-additive approximate local decomposition vector, where each iteration takes $O(m)$ time. The paper describe a fractional peeling technique to round the fractional solution for densest subgraph decompositon. The theoretical contribution is supported with experiments by comparing the proposed algorithm with some state of the art algorithms on real world and synthetic datasets.



**Questions:**

No comments for now.

**Ethics Review Area:**

["I don’t know"]

**Limitations:**

Yes.

**Strengths And Weaknesses:**

**Strengths**
- An efficient, simple, and scalable (epsilon-additive) algorithm for DSG and DSG-LD
- Fractional peeling technique with strong theoretical and empirical results
- Connection DSG-LD to decomposition of Supermodular function (DSS)
- Empirical evaluation of the described algorithms and fractional peeling technique and its comparison to the state of the art algorithms
- The algorithm being parallelizable
- The extension of the algorithm for hypergraphs
- The connection of the LP and QP for DSS

---

> ### Author Response · Authors · 2022-08-01
> **Reply for Reviewer wCmG**
>
> There were no specific questions from this reviewer at this stage. We would be happy to answer any questions from the reviewer if they come up during the discussion period.

---

### Author Response · Authors · 2022-08-01
**Message to all reviewers**

We thank the reviewers for their reviews, useful comments, and positive feedback. We appreciate the time spent on carefully reviewing the paper. We updated the main submission and the Appendix in the supplementary material. The new versions reflect the  information requested by the reviewers. Detailed responses to the specific questions and comments of each reviewer are given below.

---

### Meta-Review · Area_Chair_p6e6 · 2022-08-24

**Recommendation:** Accept
**Confidence:** Certain

**Metareview:**

Overall the reviewers found the paper to tackle an important problem and the level of technical novelty was high. The experiments were also good. Clear accept.

**Award:**

Yes

---

### Decision · Program_Chairs · 2022-09-14

Accept